# Cost-Efficient Expression of Human Cardiac Myosin Heavy Chain in C2C12 Cells with a Non-Viral Transfection Reagent

**DOI:** 10.3390/ijms25126747

**Published:** 2024-06-19

**Authors:** Albin E. Berg, Lok Priya Velayuthan, Alf Månsson, Marko Ušaj

**Affiliations:** Department of Chemistry and Biomedical Sciences, Faculty of Health and Life Sciences, Linnaeus University, 391 82 Kalmar, Sweden; albin.berg@lnu.se (A.E.B.); lok.velayuthan@lnu.se (L.P.V.)

**Keywords:** virus-free gene delivery, C2C12, cell transfection, protein expression, protein purification, human cardiac myosin II, in vitro motility assay, single-molecule assays

## Abstract

Production of functional myosin heavy chain (MHC) of striated muscle myosin II for studies of isolated proteins requires mature muscle (e.g., C2C12) cells for expression. This is important both for fundamental studies of molecular mechanisms and for investigations of deleterious diseases like cardiomyopathies due to mutations in the MHC gene (MYH7). Generally, an adenovirus vector is used for transfection, but recently we demonstrated transfection by a non-viral polymer reagent, JetPrime. Due to the rather high costs of JetPrime and for the sustainability of the virus-free expression method, access to more than one transfection reagent is important. Here, we therefore evaluate such a candidate substance, GenJet. Using the human cardiac β-myosin heavy chain (β-MHC) as a model system, we found effective transfection of C2C12 cells showing a transfection efficiency nearly as good as with the JetPrime reagent. This was achieved following a protocol developed for JetPrime because a manufacturer-recommended application protocol for GenJet to transfect cells in suspension did not perform well. We demonstrate, using in vitro motility assays and single-molecule ATP turnover assays, that the protein expressed and purified from cells transfected with the GenJet reagent is functional. The purification yields reached were slightly lower than in JetPrime-based purifications, but they were achieved at a significantly lower cost. Our results demonstrate the sustainability of the virus-free method by showing that more than one polymer-based transfection reagent can generate useful amounts of active MHC. Particularly, we suggest that GenJet, due to its current ~4-fold lower cost, is useful for applications requiring larger amounts of a given MHC variant.

## 1. Introduction

The contraction of heart and skeletal muscle is due to cyclic interactions between myosin II motor domains in the thick myosin-containing filaments and actin in the thin filaments of the ordered three-dimensional arrangement of the sarcomere. The detailed actin–myosin interaction determines the kinetic properties of muscle contraction by mechanisms that are not yet fully elucidated [1]. Accordingly, point mutations in sarcomere proteins, not the least myosin II, may have deleterious effects, but the effects of such mutations also give important information about normal function [2,3,4,5,6,7]. Point mutations in the cardiac β-myosin II motor are common causes of a hereditary disease, hypertrophic cardiomyopathy (HCM), the leading cause of sudden cardiac death in young people [8,9]. A key issue for understanding HCM pathogenesis is why the minor to moderate functional changes on the molecular and cellular levels due to the point mutations [10,11,12,13,14,15] lead to detrimental remodeling of the heart. An important question is if there is a universal mutation-induced type of functional change that is a prime stimulus for pathologic remodelling [10,12,13,14,15,16,17,18]. To test these ideas systematically, one strategy is to study the functional effects of a wide range of HCM-causing mutations under conditions where secondary remodeling effects do not blur the primary effects. For such studies, it is advantageous to use isolated proteins expressed in cell systems [19] (see also [15]) where they are not subjected to cellular forces produced by defective, mutated proteins with possible secondary remodeling effects. Moreover, such isolated proteins should ideally be of human origin due to possible inter-species differences [19,20].

Site-directed mutagenesis in cardiac myosin II is particularly suitable for studies of HCM pathogenesis. First, the gene (MYH7) for the cardiac ventricular myosin heavy chain (β-MHC) is one of the two most frequently affected genes (>30% of the cases) in HCM. Second, the functional study of mutations in β-MHC is more straightforward, both to perform and interpret than functional studies of cardiac myosin binding protein C (cMBPC), the second most frequently mutated protein in HCM (>30% of cases). This is partly because most β-MHC mutations are autosomal dominant missense mutations [11], whereas a large fraction of the cMBPC mutations are truncation mutations that both lead to underexpression (haploinsufficiency) of the entire protein and loss of various stretches of the molecule, with potentially complex consequences [11]. Moreover, the normal functional interactions between cMBPC and other sarcomere proteins are less well characterized than those of myosin.

A challenge in studying isolated heart muscle myosin containing the human β-MHC is that β-MHC is difficult to express in functional form. The only viable approach seems to be expression in mammalian muscle cells that normally express the chaperones required for correct folding of the β-MHC [21,22]. Moreover, transient rather than stable overexpression of β-MHC is required for the effective generation of proteins with different mutations. Methods to achieve this [21,22,23,24,25], employing adenovirus-based transfection of C2C12 cells, are, however, used routinely only in a limited number of labs. This may be attributed to the fact that the use of viruses for plasmid delivery makes the method both costly and time-consuming in addition to the need for access to Class 2 biosafety labs. This causes roadblocks to ideas that require the expression of a wide range of mutations, e.g., a wide range of those found to cause HCM [26], to test the possibility of universal mutation-induced functional defects. Similar studies using a wide range of mutants are also useful for elucidating basic functional aspects of the actomyosin interaction [2,3,4,5,6,7].

Recently [27], we described a virus-free transfection method that allows the expression of human β-MHC in sufficient amounts for key functional assays such as ATPase assays and in vitro motility assays. In that study, we used the transfection reagent JetPrime, a polymer that is first mixed with the DNA to be transfected followed by the addition of the mixture to the cells. To the best of our knowledge, there is no previous report of the use of any other non-viral transfection reagents for the expression of human β-MHC in C2C12 cells. In a recent study, we demonstrated that the JetPrime-based expression system can be used economically and effectively to express and purify human cardiac β-myosin for basic biochemical characterization. We thus found that ng quantities of myosin from one 60 mm cell cultivation plate of C2C12 cells are sufficient to derive both basal and actin-activated myosin ATPase activities using single-molecule fluorescence-based analysis [28]. We also argued that the method may be further developed to characterize a range of transient kinetics events using similar minimal amounts of protein. However, other assays for myosin characterization require more protein. This includes ultrastructural analysis [25] and in vitro motility assays with fluorescence microscopy-based observation of actin filament propulsion by surface-immobilized myosin motor fragments [24,27]. It would therefore be of interest to investigate if other, less expensive non-viral transfection agents than JetPrime may be useful for transfecting β-MHC to C2C12 cells. The existence of more than one useful transfection agent is also essential for the sustainability of the approach in case the useful agent for some reason becomes obsolete.

Based on a web search for any non-viral commercial cell transfection reagents with available manufacturer application notes for C2C12 cells we identified GenJet (SignaGen Laboratories, Frederick, MD, USA) reagent as a suitable alternative to JetPrime. Similar to JetPrime, it is a polymer-based transfection reagent but with a significantly (~4.5×) lower price per volume (as estimated in April 2024). Here, we have assessed the transfection efficiency of the expression plasmid for an 848 amino-acid-long human β-MHC motor domain construct fused to an enhanced green fluorescent protein (eGFP) using GenJet as a transfection reagent. Further, we have determined the protein purification yields and verified the enzymatic activity of the construct using an in vitro motility assay and single-molecule fluorescence-based ATPase assays [28,29]. We show that GenJet exhibits a transfection efficiency approaching that of JetPrime if our developed and optimized transfection protocol is used. Similarly, the protein purification yields are slightly lower than those after transfection with JetPrime. Nevertheless, functional assays confirmed that the purified myosin construct is active. Considering the almost four-fold lower cost of GenJet compared to JetPrime when taking into account the slightly lower GenJet expression and purification yields, our results suggest that GenJet is a recommendable reagent for large-scale vertebrate muscle myosin production in C2C12 cells.

## 2. Results

We used the transfection reagent GenJet to transfect C2C12 cells and compared its efficiency to our recently described reagent JetPrime [27], which served as the positive control for this study. Similar to the JetPrime transfection reagent, GenJet is accompanied by a protocol (application note) for C2C12 cells. However, further protocol development tailored to our needs was required (see below). All transfection experiments were performed using our working plasmid, carrying the human β-cardiac myosin motor domain construct (S1L) described in detail recently [27], as well as below (Section 4). This construct, codon optimized for expression in host cell lines of mouse origin, is fused to enhanced green fluorescent protein (eGFP) with a FLAG tag at its C-terminal. As with any polymer-based cell transfection reagent, the mode of action relies on properly formed DNA:GenJet complexes. For all the transfections, we have kept the ratio of DNA:GenJet (*w*/*w*) constant (1:4) as recommended in the manufacturer application note.

### 2.1. Transfection of Cells in Suspension

Because the manufacturer’s application note for C2C12 cells suggests the transfection of the cells in suspension, we first tested that approach. By following the protocol, we transfected 1.2 million C2C12 myoblasts in suspension, after which the cells were seeded onto 35 mm cell culture dish for further differentiation into myotubes. The first observation was that the differentiation process was severely impaired as can be seen from a substantial number of rounded cells suggesting an inability to form elongated myotubes (Figure 1, brightfield images). An attempt to improve the differentiation process by increasing the cell number to 2 million did not solve the issue (Figure 1). Moreover, regardless of the cell number used, the fluorescence microscopy imaging showed a low transfection efficiency indicated by only a few fluorescent (small) myotubes with overall weak fluorescence intensity (Figure 1). We thus abandoned this approach and proceeded with the transfection of adherent cells by combining the manufacturer suggested protocol for transfecting C2C12 cells in suspension with our previous protocol [27] for transfecting adherent and confluent C2C12 myoblasts using JetPrime.

### 2.2. Transfection of Surface-Adherent Cells

The most optimal approach for vertebrate muscle myosin production would be to deliver expression plasmid into fully differentiated myotubes since the folding machinery is already fully available, in contrast to undifferentiated myoblasts. This is indeed an approach described in most of the literature where expression plasmids are delivered by viral infections [23,24,25,30,31,32,33]. Encouraged also by personal communication with a GenJet manufacturer representative, we then tried applying GenJet to transfect fully differentiated C2C12 myotubes. However, to our disappointment, but not completely unexpectedly, we found that transfection of already differentiated cells did not seem feasible (Figure 1).

We then proceeded with our recently developed protocol by transfecting confluent myoblasts following their immediate transition to differentiation media for myotube formation [27]. In search of the most optimal combination of GenJet with the transfection of adherent and confluent myoblasts, several conditions were tested (Table 1; see also Section 4 for details).

The first clue for a successful transfection was found (Figure 1) by simply adding the same volume and amount of prepared transfection complexes as would be used for the transfection of the cells in suspension to the adherent and confluent cells in a 35 mm dish (condition GJ-1 in Table 1). Reasonably good observed transfection efficiency demonstrated the decent capability of the GenJet reagent to be used on adherent and confluent C2C12 myoblasts. To further improve transfection efficiency, we tested other conditions (Table 2). Our major concern regarding the condition GJ-1 was that the transfection solution volume was rather low (~200 µL), possibly insufficient to fully cover the adherent cell monolayer in a 35 mm dish. Thus, we first doubled the volumes, including those of DNA and GenJet (condition GJ-2, Table 1). This resulted in a better transfection efficiency approaching that of JetPrime (Figure 1). However, doubling the amount of DNA and GenJet downgrades the promised cost efficiency of the proposed method. Thus, we downscaled the DNA and GenJet volumes to the original proposed one in condition GJ-3 (Table 1). The formation of DNA:GenJet complexes was conducted in the original recommended volume so as not to disturb the process by overdiluting the DNA and GenJet. However, the total transfection volume was doubled by pre-adding the same amount of solution for complex formation (DMEM) to the cells intended for transfection (condition GJ-3, Table 1). This approach inevitably diluted the original [DNA:GenJet] transfection complex concentration by half for cell transfection. To offset this presumably negative consequence to transfection efficiency, we designed and tested in parallel a variation of condition 3 by doubling the transfection time to 40 min. As can be seen from representative images in Figure 1 good transfection efficiency approaching that of JetPrime was achieved by either variation of condition GJ-3.

Quantitative data for transfection efficiency based on fluorescence microscopy imaging (Figure 1B), as well as Western blot analysis of total cell lysates (Figure 2), for different transfection conditions (Table 1) support these conclusions. In these analyses, we have compared the outcomes of different GenJet transfection conditions with JetPrime transfection performed using a previously JetPrime-based optimized protocol as a positive control (PC). From these experiments, the major discovery was that doubling the total volume of transfection solution without changing DNA and GenJet quantities (conditions GJ-1 vs. GJ-3 in Table 1) improved the transfection efficiency. We attributed the effect to the higher volume per se (i.e., better covering of the cell monolayer), as suggested by the comparison of conditions GJ-1 and GJ-3. In condition GJ-3 the volume was similar to condition GJ-2 (Table 1) however the concentration of the transfection complexes and, thereby, their total amount was halved compared to condition GJ-2. Yet, the transfection efficiency was similar under conditions GJ-2 and GJ-3, suggested both by image analysis of the cells (Figure 1B) and by densitometric analysis of Western blots of the cell lysates (Figure 2). An increased incubation time from 20 to 40 min did not increase the efficiency, arguing against limitations associated with the rate of the transport of the transfection complexes to the cells. The similarity of the data for conditions GJ-2 and GJ-3, including different incubation times in the latter case, motivates the pooling of these data (the most rightward data points in Figure 1B and Figure 2B).

Overall, the comparison of GenJet and JetPrime transfection shows that the transfection efficiency using GenJet is on average ~15% lower than that with JetPrime based on fluorescence microscopy image analysis (Figure 1B). Furthermore, analysis of the Western blots of the total cell lysates in Figure 2B suggests 28 ± 3% (mean ± 95%; assuming normal distribution) lower protein production with GenJet relative to the ones transfected with JetPrime. The GenJet transfection condition GJ-3/20min was then chosen for further experiments.

Next, we determined the optimal cell harvesting time after transfection for the high protein expression (Figure 3). The expression increased over time in terms of total transfected cell area, as well as the amount of expressed protein, assessed by Western blot analysis, showing a tendency towards saturation at days 7–8, like our previous report using JetPrime [27]. We have chosen day 7 as a standard cell harvesting day for protein purification, also to be consistent with our and other previous reports [27,34].

### 2.3. Protein Purification

The above optimization is the basis for our final protocol and its adaptation for expression, purification, and functional characterization of a β-cardiac myosin S1L construct (S1L-eGFP-FLAG; usually S1L below). The purification steps are described in detail in the Section 4, as well as in our previous paper [27]. After harvesting the cells on day 7 post-transfection, the cells were lysed, and the S1L construct was captured using a purification resin (Figure 4A, top) consisting of micrometer-scale beads decorated with anti-FLAG antibodies. The bound protein was eluted by competition with excess of free FLAG peptide (Figure 4A, bottom). A representative Western blot on purified myosin preparations from cells grown in a 60 mm cell culture dish is shown in Figure 4B. The densitometry analysis showed that protein purification yield from cells transfected with GenJet was ~22% lower compared to the yield achieved with cells transfected with JetPrime. This is in line with the assessment of the transfection and expression efficiency analyzed above (Figure 1 and Figure 2). Similarly, estimated purification yields of 0.25 ± 0.22 µg (n = 3; mean ± SD) for GenJet and 0.27 ± 0.06 µg (n = 3) for JetPrime from one 60 mm cell culture dish were achieved.

### 2.4. Functional Assays

We next proceeded to characterize the enzymatic activity of the purified myosin construct. Due to recent advancements in our single-molecule fluorescence-based ATPase assays [28,29], we utilized them to determine purified myosin activity in terms of its basal (*k*_basal_) and actin-activated (*k*_cat_) steady-state ATP turnover rate constants. The details about the assays can be found in the aforementioned publications and below (Section 4). The rate constants *k*_basal_ and *k*_cat_ were estimated from the binding kinetics of fluorescent ATP analog (Alexa647–ATP) to purified S1L, either immobilized via antibodies to a surface (Figure 5A,B; *k*_basal_) or cross-linked to surface-immobilized actin filaments (Figure 5C,D; *k*_cat_). The steady-state ATPase activities were estimated from fits to cumulative on-time distribution data (Figure 5E–H), with kinetic parameters summarized in Table 2. When comparing the extracted rate constants and amplitudes, no differences were observed between proteins purified from cells transfected with GenJet or JetPrime. Cross-linking purified myosin construct molecules with actin filaments increased the steady-state ATPase activity of S1L-eGFP-FLAG more than 100-fold (from *k*_basal_~0.05 to *k*_cat_~6.4 s^−1^) similar to what was found in our previous publications [27,28]. Similarly, an unspecific ATP binding process (*k*_unspecific_ ≈ 0.3–0.5 s^−1^) [29] was also found in both GenJet and JetPrime conditions without apparent differences in its rate constant and amplitude (Table 2).

To fully appreciate and further examine the chemomechanical transduction cycle, we next performed an in vitro motility assay (IVMA) where actin filament gliding by surface attached S1L is observed. The IVMA is also a more sensitive assay regarding the fraction of active heads in the purified S1L-eGFP-FLAG sample. The IVMA results are presented in Figure 6 (Appendix A). Actin filament gliding velocities were on average 1.2 µm/s (Table 2). The fraction of motile filaments (FMF) reached an average of 60% (Table 2). As in previous studies (Ref. [27] and refs therein), removal of inactive heads by affinity purification (“deadheading”) was needed to achieve good motility. For the myosin construct, purified from cells transfected using GenJet, double “deadheading” was necessary to observe motile filaments. This suggests a relatively high number of inactive heads in the original protein elution sample. The measurements of protein concentrations via eGFP fluorescence before (300 nM) and after double-deadheading (125 nM) support that suggestion. The observed loss is consistent with a fraction of inactive heads of about 50% as suggested by observed spots without co-localization of Alexa 647-ATP and eGFP fluorescence in Figure 5A. The findings suggest that further rounds of deadheading would increase both velocity and the fraction of motile filaments further.

The assay with myosin cross-linked to actin is designed to measure *k*_cat_ as elaborated on recently [28] and we have not yet developed a single-molecule assay to assess the actin affinity. Whereas it can be measured using solution-based assays at different actin concentrations (e.g., [27]), we have not applied that approach here. However, the in vitro motility assay experiments suggest that the average actin affinity to myosin, produced with GenJet, is reasonably consistent with results obtained using JetPrime [27] or virus-based DNA-delivery [24]. If the affinity had been very much lowered, we would have also seen rather long filaments diffusing back and forth close to the surface rather than moving steadily forward in the presence of methylcellulose as used here. If, on the other hand, the affinity had been very high, the motility would be greatly inhibited with a large fraction of stationary filaments and very low velocity.

### 2.5. Cost Analysis of Transfection Reagents: GenJet vs. JetPrime

The substantially lower shelf price of GenJet in comparison to JetPrime was the main reason to explore the efficacy of the GenJet transfection reagent to produce the myosin motor proteins in C2C12 cells. However, besides the volumetric price of the reagent (price/µL, Table 3), one should also consider other factors. One of them is the amount of reagent used per transfection. This is further dependent on dish size, and it does not necessarily scale linearly. Thus, the price/performance ratio was calculated based on the most used dish in our protein purification experiments: the 60 mm cell culture dish (price/60 mm dish, Table 3). Finally, the protein expression and purification yield per certain number of cells (or cell culture dish) needs to be considered to justify the comparison. Here, we have different estimations available (Figure 1, Figure 2 and Figure 3) all in favor of JetPrime. We took the most conservative estimation of up to 30% lower protein yield in the case of GenJet transfection, which means up to 43% more cells need to be grown for the same amount of S1L-eGFP-FLAG protein. The transfection reagent GenJet is thus at least 3.6 times more cost-effective than JetPrime (Table 3) to produce the same amount of S1L-eGFP-FLAG myosin construct under our optimized experimental conditions.

## 3. Discussion

GenJet™ In Vitro DNA Transfection Reagent (Ver. II) has been previously used for expressing constructs such as Renilla Luciferase (to easily monitor and quantify the transfection) in a wide range of cells such HEK293, HeLa, Cos-1, CHO, and NIH 3T3 cells (manufacturer website: https://signagen.com/In-Vitro-DNA-Transfection-Reagents/SL100489/GenJet-In-Vitro-DNA-Transfection-Reagent-Version-II (accessed on 1 June 2024)). Also, other constructs have been expressed in different cells previously. EGFP has been expressed in mouse embryonic fibroblasts [35]. Also, Arc (activity-regulated cytoskeleton-associated protein, 48 kDa) has been successfully expressed in living U2OS cells using GenJet [36]). Furthermore, the GenJet transfection reagent has been used to generate stable HeLa cell lines [37]. Finally, IGFN1 (Immunoglobulin-Like And Fibronectin Type III Domain-Containing 1) was expressed in C2C12 cells [38]. Despite having been previously used to also transfect C2C12 myoblasts in suspension, we demonstrate here that the GenJet transfection reagent is also useful for transfecting C2C12 myoblasts growing on a surface. For that purpose, we used a protocol modified from that described for cell transfection in suspension while also mimicking key steps in the JetPrime-based transfection that we described previously [27]. We also find that the myoblasts after transfection with the plasmid carrying the S1L-eGFP-FLAG myosin construct can be differentiated into myotubes while expressing the protein in quantity (≥70%) approaching that of JetPrime. We did not investigate the basis for somewhat lower expression and purification yields in detail but different factors may contribute, from the differences in DNA:polymer complex properties (e.g., size) to the differences in efficiencies of endocytosis, transport to the nucleus, endosomal escape, and final plasmid release (complex disassembly) to enable translation [39,40]. Following the time course of the transfected cell area (Figure 3) and comparing it to that of JetPrime [27], we found that the expression expansion is delayed for about 1–2 days. This suggests that 8–9 days post-transfection would be needed to reach the same expression yield as JetPrime (see Appendix A). However, keeping transfected cells beyond day 7 under our experimental conditions dramatically increased the risk thatthe cell monolayer detached from the surface to clump together into a visible cell aggregate with unknown consequences to the expressed myosin construct activity. The reason for such detachment of transfected cell monolayer (regardless of transfection reagent used) in comparison to non-transfected cells which we could keep in the differentiation media way beyond day 7 (up to day 19, where we could trigger beating pattern after external electric stimulation; see Appendix A) remains to be solved in the future. This may have the potential to increase the GenJet-based protein expression level to the level seen with JetPrime. The reasons for expression delay could be multifaced; however, one could argue that there was initially a lower number of transfected cells on day one with GenJet (1.6%, Figure 3) in comparison to JetPrime (4.6%, see ref. [27] and Appendix A). This should not come as a complete surprise as we move away from the official procedure found in the manufacturer’s manual (transfection of cells in suspension). For example, following the official manual, the transfection complexes are not removed at any step, merely diluted with growth media after 20 min. In our optimized protocol, which includes efficient myoblast differentiation to myotubes after transfection, the complexes are completely removed 3.5 h post-transfection, when cells are transferred to differentiation media. One could add additional transfection complexes to the differentiation media. However, that would severely diminish cost efficiency and may increase intracellular stress.

The internal cell state (e.g., stress levels, viabile status, etc.) can also contribute to the final protein yield. We have observed that JetPrime transfection is usually associated with denser cell monolayers occasionally even areas with cell multilayers are observed on the day of harvesting. This could potentially suggest that GenJet is more cytotoxic than JetPrime. The issue may be clarified with proper cell viability assays, but the lower S1L-eGFP-FLAG expression level by up to 28% is not critical for our purposes, so we decided not to pursue the issue.

The expressed protein was successfully purified and we have shown, using recently developed single-molecule fluorescence ATPase assays [27,29] and the in vitro motility assay, that the GenJet-expressed S1L-eGFP-FLAG myosin construct is functional. The basal (*k*_basal_) and maximal actin-activated ATP turnover (*k*_cat_) rate constants (Table 2) are in agreement with what was previously found for the same construct using solution (NADH-coupled) assay (~0.04 s^−1^ and ~8 s^−1^ as reported in [27]) and single-molecule ATPase assays (~0.03 s^−1^ and ~6.8 s^−1^ published in [28]). We have explained the small difference in rate constant values of the fast phase (*k*_cat_) when comparing the one determined using the solution (NADH-coupled) assay with the ones originating from single molecule assays by the current limitation in time resolution (i.e., 50 ms) of our camera [28].

The gliding velocity and fraction of motile filaments were somewhat lower than previously found using either JetPrime (Table 2) or virus-based gene delivery considering the same length of the myosin construct (Ref. [27] and references therein). This is most straightforwardly explained if larger amounts of non-functional myosin motor heads are present in protein preparation from cells transfected with GenJet because even a small proportion of inactive motors can slow velocities noticeably [24]. The necessity of double affinity purification (“deadheading”) to observe motile filaments (in contrast to the only one needed in previous JetPrime-based expression) and purification experiments [27], as well the estimated fraction of protein remaining in the supernatant after double-deadheading (~40%), support this implication. The presence of non-functional recombinant myosins in protein preparations is a usual phenomenon for such hard-to-express enzymes. The determination of a fraction of active heads via, e.g., single turnover of mant-ATP in a stopped-flow apparatus [41] or their removal via “deadheading”, described here, are typical next steps after protein elution from purification beads. However, the fractions of active protein obtained are usually not disclosed in the publications. Similarly, we could not find any quantitative data relating the exact number of “deadheadings” required to observe motile filaments coupled with the fraction of motile filaments in previous works using a similar myosin construct obtained with viral gene delivery for comparison. The removal of inactive motors was performed multiple times until all filaments were moving smoothly [24] or until the percentage of stuck filaments dropped below 10% [31,42]. It seems likely that while using myosin motor constructs purified from cells transfected by GenJet reagent, a similar approach of several “deadheadings” would most likely bring the observed actin filament velocities closer to the values (1.5–1.8 µm/s) reported in our previous study and previous publications by others (Ref. [27] and references within). Importantly, therefore, we do not believe that the lower velocity and possibly higher level of dead heads is a general characteristic associated with GenJet. Rather, there seems to be a variability between preparations for any transfection/purification method as indicated by the non-specified multiple removals of inactive motors [24,31,42]. Another possibility that is less likely but that we cannot exclude based on our available data is that reduced or different co-purification of essential or regulatory light chains (cf. [43,44,45]) contributes to the lower velocity with GenJet transfection.

GenJet shares similar advantages with JetPrime over the virus-based gene delivery in reduced cost and time consumption for the production of isolated striated muscle myosin II. Moreover, these non-viral approaches avoid the need for safety labs for virus handling. The benefits concerning time consumption and cost are particularly appreciable if a study requires small amounts of myosin only and the generation of a wide range of different mutants when each mutant requires the generation of new viruses if the virus-based system is used. Due to the at least 3.6 times better price/performance ratio of GenJet to JetPrime, GenJet has the potential to be particularly useful in substituting JetPrime when performing large-scale cell transfection for substantial protein quantities. Moreover, another advantage of GenJet lies in the fact that there is no need for a proprietary buffer solution (with unknown composition) for DNA complex formation. Rather this is achieved in regular DMEM, offering more flexibility in conducting the experiments. Furthermore, less DNA per GenJet transfection is needed in comparison to JetPrime (e.g., 5 µg vs. 11.25 µg for 60 mm cell cultured dish), which further improves cost efficiency. On the other hand, due to slightly lower transfection and expression efficiency and probably more non-functional heads, JetPrime is still our transfection reagent of choice when small amounts of myosin are to be expressed, such as for single-molecule assays [28]. It would also be interesting to investigate the possibility of using either JetPrime and/or GenJet in future studies for expression of longer heavy meromyosin-like constructs as previously achieved using adenovirus-based DNA-delivery [46]. This would allow studies to give a wider range of insights, e.g., also related to interactions between the two heads of a given myosin and the effects of mutations on these interactions [46].

GenJet also shares similar disadvantages with JetPrime when it comes to the production of considerably higher amounts of protein. The amount of S1L produced here and in our previous works [27,28] would, for instance, not be effectively upscaled to the amounts required for conventional solution-based transient kinetics [47] or ultrastructural studies [25]. However, importantly, we have shown recently that <1/10 of the S1L produced by the C2C12 cells in a single 60 mm cell culture dish is required to measure both basal and maximum actin-activated myosin ATPase in single-molecule studies [28]. Of note, the determination of those rate constants using single-molecule assays is not limited by the possible presence of non-functional heads. Even more, knowledge of the concentration of active heads is not needed for their calculations in contrast to solution kinetic techniques (e.g., cuvette-based NADH-coupled assay [27]). We expect to expand these single-molecule assays to various transient kinetics phenomena (see [28]). For assays requiring intermediate amounts of protein and that give different information than the single-molecule methods, such as in vitro motility assays, the use of non-viral transfection is an option as demonstrated here and previously [27]. Other applications will most likely require virus-based expression such as ultrastructural studies (e.g., X-ray crystallography and cryo-electron microscopy) due to appreciable quantities of proteins needed. One may wonder if higher expression efficiencies in terms of active protein could possibly be obtained using the non-viral methods or if the maximum has been reached [27]. The fact that purification yields using either the JetPrime or GenJet reagents are in the same range (i.e., <0.1 µg/cm^2^ of cell cultivation area), and further that similar transfection efficiency was obtained under a range of conditions using GenJet, seems to support the idea that saturation is reached. However, the polymer-based transfection efficiency can be improved if aided by other means. For example, it was shown that polyethyleneimine (PEI)-based transfection of adherent cells can be increased by up to 100-fold employing milli-to-nanoscale vibrational cues in the so-called vibropolyfection procedure [48]. It would be interesting to explore this and similar possibilities to improve the transfection efficiency of GenJet (and JetPrime) in the future.

## 4. Materials and Methods

### 4.1. Cells, Chemicals, and Materials

C2C12 mouse myoblasts (ATCC CRL 1772) were purchased from Sigma (Sigma-Aldrich, Germany, now Merck (Darmstadt, Germany)). Dulbecco’s Modified Eagle Medium (DMEM, cat. no. 01-055-1A) was purchased from Sartorius (Göttingen, Germany). Fetal bovine serum (FBS, HyClone, Cat. no. SH30071.03), antibiotic/antimycotic solution (HyClone, cat. no. SV30079.01), horse serum (HyClone, cat. no. SH30074.02) were from Cytiva (Marlborough, MA, USA). GlutaMAX (cat. no. 35050061), non-essential amino acids (NEAA, cat. no. 11140050), Dulbecco’s phosphate-buffered saline (DPBS, cat. no. 14190250) and 1-ethyl-3-(3-dimethylaminopropyl) carbodiimide hydrochloride (EDC, Cat. no. 22980) were from Gibco (Thermo Fisher Scientific, Waltham, MA, USA).

Insulin (cat. no. I9278), 0.25% Trypsin-EDTA solution (cat. no. T4049), HEPES (cat. no. H4034), 2-(N-Morpholino) ethanesulfonic acid hydrate (4-morpholineethane-sulfonic acid (MES Hydrate, cat. no. M8250), pyranose oxidase (POX, cat. no. P4234), cyclooctatetraene (COT, cat. no. 138924), 4-nitrobenzyl alcohol (NBA, cat. no. N12821), Trolox (cat. no. 238813), creatine phosphate (CP, cat. no. P7936), creatine phosphokinase (CPK, cat. no. C3755), bovine serum albumin of high purity (BSA, cat. no. A0281), adenosine 5′-triphosphate disodium salt hydrate (ATP, cat. no. A23A3), phenylmethanesulfonyl fluoride (PMSF, cat. no. P7626), imidazole-HCl (cat. no. I3386) were from Sigma (Sigma-Aldrich, Germany, now Merck).

All other chemicals were of analytical grade and purchased from Sigma (Sigma-Aldrich, Germany, now Merck) unless stated otherwise below.

### 4.2. Ethical Statement

The rabbits were euthanized to purify actin by following ethical procedures approved by the Regional Ethical Committee for Animal Experiments in Linköping, Sweden (ref. no 17,088–2020). All the procedures were performed by the veterinarian at Linnaeus University. For euthanasia, the rabbit was initially anesthetized by injecting 0.25 mL Zoletil (active ingredients: Zolazepam, 6 mg/kg; Tiletamine, 6 mg/kg, and Medetomidine, 0.6 mg/kg) intramuscularly followed by another injection in the ear vein with 2 mL Pentobarbital (100 mg/mL). Dorsal lateral back muscle was extracted immediately after euthanasia and actin was purified following the protocols in [49].

### 4.3. Plasmid Design

The plasmid construct used to express myosin is described in detail in our previous publication [27] and is depicted in Figure 7. The plasmid was engineered and purchased from GenScript Biotech Corporation. The MYH7 gene from https://www.uniprot.org/uniprot/P12883 (accessed on 1 June 2024) was truncated and codon-optimized to express 1-848 amino acids that code for the heavy chain of human cardiac β-myosin motor domain (subfragment 1; S1; here S1L), along with the binding sites for essential light chain (ELC) and regulatory light chain (RLC). The S1L sequence was fused to TEV (tobacco etch virus) protease site followed by enhanced green fluorescent protein (eGFP) and a FLAG tag. This gene sequence was then integrated into the pcDNA-3.1 vector backbone under the CMV promoter.

### 4.4. Production of Plasmid

Plasmid stock was prepared by introducing the circular plasmid (pcDNA-S1L-eGFP-FLAG) into competent *E. coli* (DH5-α) cells. The transformed bacteria were grown overnight on Luria–Bertani (LB) agar plate containing 100 µg/mL ampicillin. A single bacterial colony was inoculated into 10 mL LB liquid medium supplemented with ampicillin (starter culture) and was grown for 7–8 h in a shaking incubator at 37 °C, 170 RPM (revolutions per minute). The starter culture was diluted 1/1000 times and was grown in larger volumes (500 mL) supplemented with ampicillin overnight. Cells were harvested the next day for plasmid extraction and purification according to the manufacturer’s protocol using QIAGEN Plasmid Maxi Kit (Qiagen, Hilden, Germany). The purified plasmid pellet was reconstituted in MilliQ water to obtain a concentration of ~1 µg/µL for transfecting C2C12 cells either with JetPrime or GenJet reagents. The plasmid concentration was determined using the NanoDrop ND-1000 UV-Vis Spectrophotometer (ThermoFisher Scientific, Waltham, MA, USA).

### 4.5. Routine C2C12 Cell Handling

A vial containing 1 million C2C12 cells was thawed for 1–2 min in a water bath at 37 °C. The C2C12 cells were then transferred to a T175 flask with 25 mL growth medium containing DMEM supplemented with 10% FBS, 1% GlutaMAX, and 1% antibiotic/antimycotic. The cells were placed in a humidified incubator at 37 °C supplied with 5% CO_2_ (Forma Series II 3110 Water-Jacketed CO_2_ Incubator, Thermo Fisher). Following ~6 h incubation, the medium was exchanged with fresh growth medium (20 mL), and the cells were allowed to grow until 60–70% confluence.

After reaching the desired confluence, cells were sub-cultured by removing the growth medium, followed by rinsing the cells with DPBS. Pre-warmed trypsin was added to the cells in volumes corresponding to half of the initial growth medium volume and incubated at room temperature (RT; 21–23 °C) for 1–2 min. To dislodge the cells completely from the surface, ~80% of the trypsin solution was removed and the cells were placed inside the incubator for an additional 1–2 min followed by gentle tapping of the flask. Trypsin was neutralized by adding growth medium approximately 5 times the volume of the remaining trypsin solution. A volume of 10 µL cell suspension was transferred to a hemocytometer for cell counting. The volume, V, of the cell-containing solution in mL, to be transferred to a new cell flask for new cell passage was calculated as:(1)$$V=\frac{\left(\mathrm{desired}\;\mathrm{number}\;\mathrm{of}\;\mathrm{cells}\times 10^{3}\;\mathrm{per}\;{\mathrm{cm}}^{2}\right)\times \mathrm{flask}\;\mathrm{surface}\;\mathrm{area}\;\mathrm{in}\;{\mathrm{cm}}^{2}}{\mathrm{average}\;\mathrm{number}\;\mathrm{of}\;\mathrm{cells}\;\mathrm{counted}\;\mathrm{in}\;\mathrm{hemocytometer}\times 10^{4}\;\mathrm{per}\;\mathrm{mL}}$$

The desired number of cells seeded to the new culture flask was determined by following Table 4.

### 4.6. Cell Seeding before Transfection

Cells were seeded in 3 × 10^5^ or 6 × 10^5^ in 35 mm or 60 mm culture dishes (Nunclon™ Delta cat. 153066 or Sarstedt culture dish cat. no. 833901, respectively) in 2 or 5 mL growth medium according to a previously established routine [27]. The cells were transfected when the confluence reached ~95% (~36 h after seeding). For experiments run in parallel, 6-well multiwell plates were also used (Sarstedt, Nümbrecht, Germany).

### 4.7. Cell Transfection Using JetPrime

For the preparation of the transfection complex for one 35 mm cell culture dish, 3 µL (1 µg/µL) plasmid DNA was mixed with 200 µL JetPrime buffer. JetPrime reagent of 6 µL (1:2, DNA:JetPrime volume ratio) was added followed by a quick vortex and spin down. For a 60 mm dish, the final complex formation volume was 500 µL containing 11.25 µL plasmid and 22.5 µL transfection reagent in JetPrime buffer. The mixture was incubated at RT for 10 min to allow the formation of a transfection complex. The transfection mix was added dropwise to the growing cells followed by incubation at 37 °C for 3.5 h. The transfection process was halted by removing the growth medium, washing the cells once with DPBS, and replacing it with differentiation medium for myogenic differentiation. The differentiation medium contained DMEM, 2% horse serum, 1% GlutaMAX, 1% antibiotic/antimycotic, 1% NEAA, 1 µM insulin, and 25 mM HEPES. The differentiation media was exchanged every 24 h. The cells were harvested on day 7 after transfection.

### 4.8. Cell Transfection Using GenJet

#### 4.8.1. Preparation of Transfection Complex

For cell transfection using GenJet reagent, we first followed recommended manufacturer application manuals, where cells were transfected in suspension. For a 35 mm dish (or one well of a 6-well multiwell plate), the transfection complexes were formed by mixing 2 µL plasmid DNA (1 µg/µL), 8 µL GenJet reagent (1:4 ratio), and 200 µL serum-free DMEM supplemented with 1% antibiotic/antimycotic, followed by brief vortexing and incubation at RT for 15 min (see also Table 1 in Results section). For a 60 mm dish, the complex formation volume was ~500 µL containing 5 µL plasmid (1 µg/µL) and 20 µL transfection reagent (last row in Table 1). Note that regardless of the cells being transfected in suspension or adherent mode (see below), the complexes were prepared using the same approach.

#### 4.8.2. Transfection of Cells in Suspension

The manufacturer’s application note [50] for C2C12 cell transfection suggests performing transfection on cells in suspension. The original protocol suggests using 1.2 × 10^6^ myoblasts to be transfected for one 35 mm dish (or one well of a 6-well multiwell plate). This did not result in good differentiation to myotubes in the days post-transfection. Trying to ameliorate this undesired outcome, an increased number of 2 × 10^6^ C2C12 cells were collected from a routine cell culture flask and centrifuged at 160× *g* for 10 min. The supernatant containing the complete growth medium and trypsin was removed, and the cells were washed with 2.5 mL DPBS. Centrifugation was repeated once again with the same conditions and the cell pellet was resuspended in transfection complex solution as described above and incubated at 37 °C in a water bath for 20 min.

#### 4.8.3. Transfection of Adherent Cells

As mentioned above, it is important for our purpose that the C2C12 myoblasts undergo immediate differentiation into myotubes after transfection. This is most readily achieved by transfecting the confluent and adherent culture of C2C12 myoblasts. We, therefore, combined the protocol for transfection in suspension described above with our previous protocol [27] for transfection of adherent and confluent C2C12 myoblasts using JetPrime. C2C12 cells were seeded and grown for two days until ~95% confluence was achieved. The growth medium was removed and the solution with the transfection complexes (see Results Section Table 1 for different conditions) was added dropwise to the cells. The cells were then incubated at 37 °C for 20 min. Note that in the final optimized protocol, the same volume of DMEM (i.e., 200 µL for 35 mm and 500 µL for 60 mm cell culture dish) was pre-added to the cells before the addition of transfection complexes. After incubation, prewarmed growth medium (2 or 5 mL for 35 mm or 60 mm cell culture dishes, respectively) was added and the cells were incubated for another 3.5 h after which they were rinsed with 2 or 5 mL DPBS followed by replacement with differentiation medium (2 or 5 mL). During the differentiation process, the differentiation medium was exchanged every 24 h up to 8 days.

#### 4.8.4. Transfection of Fully Differentiated Myotubes

The procedure to transfect fully differentiated C2C12 cells was identical to that described in the paragraph above for transfection of adherent myoblasts. However, in this case, the cells were left to differentiate into myotubes before transfection for 5 days while the transfection efficiency was assessed at 7 days post-transfection.

### 4.9. Harvesting and Subsequent Storage of Transfected C2C12 Cells

The C2C12 cells were harvested on day 7 after transfection. The differentiation medium was first removed followed by washing the cells using DPBS (once). The cells were then harvested using a cell scraper (VWR Cell Lifter, cat. no. 76036-006), collected by a cut pipette tip, and transferred to an Eppendorf tube which was centrifuged at 500× *g* for 5 min. The supernatant was discarded, and the cell pellet was snap-frozen in liquid nitrogen, followed by storage in a −80 °C freezer.

### 4.10. Fluorescence Microscopy with Image Processing

The cells were observed under a fluorescence microscope (Axio Observer D1, Zeiss, Oberkochen, Germany) equipped with a mercury lamp (HBO 103 W/2, Osram, Muenchen, Germany) using a FITC filter set and 10× objective to assess transfection efficiency based on eGFP fluorescence. Images were acquired by randomly selecting 5 regions of interest in the cultivation dish using an EMCCD digital camera (C9100-12, Hamamatsu photonics, Hamamatsu-shi, Japan) attached to the microscope and controlled by dedicated HCImage software (ver. 4.51). Unless stated otherwise, the exposure time (180 ms), gain (100), and image depth (8-bit) were kept constant. To maintain low light intensity, a discrete attenuator (part number 423,647, manufactured by Zeiss) was strategically set at level 5. In this setting, the transmission is approximately 20%. The primary purpose of this attenuator was to prevent excessive exposure of the cells to excitation light, thus minimizing the risk of phototoxicity. The transfected cell area, as well as the total fluorescence intensity, was estimated using Fiji (ver. 1.54f) script as before [27]. The product of the transfected area and the total fluorescence intensity (in arbitrary units) is taken as the measure of transfection efficiency (TE). Images were acquired in different intervals post-transfection up to 8 days. If needed, image brightness and contrast were adjusted in Fiji (ver. 1.54f) [51] (Image/Adjust/Brightness/Contrast).

### 4.11. Total Cell Lysate Preparation

Cells harvested from one 35 mm plate were lysed on ice for 30 min by addition of 320 µL RIPA/lysis buffer containing 150 mM NaCl, 5 mM EGTA, 1% Triton X-100, 0.1% SDS, 25 mM Tris–HCl, 1× complete mini protease inhibitor cocktail (Roche, Ref: 11836170001). After incubation on ice, the cells were placed in a sonication bath (Branson 2510-DTH Ultrasonic Cleaner) for 30 s in RT to further increase lysis and reduce the viscosity of the cell lysate. The cell lysate was centrifuged (Eppendorf, 5430R, Hamburg, Germany) at 17,949× *g* (13,000 RPM) for 10 min at 4 °C and the supernatant was used to run SDS gel electrophoresis.

### 4.12. SDS-PAGE and Western Blot

A 20 µL sample volume was mixed with sample buffer (Pierce LDS non-reducing sample buffer, ThermoFisher, Ref: 8478) and 50 mM dithiothreitol (DTT). The mixture was heated at 95 °C for 5 min and the samples were loaded onto NuPAGE 4–12% Bis-Tris gel (ThermoFisher, Ref: NP0322BOX). The gel was placed in 400 mL NuPAGE MES SDS Running buffer (ThermoFisher, Ref: NP0002) and electrophoresis was performed at 90 V for 35–40 min, followed by 140 V for 1–1.5 h in a cold room.

For Western blot analysis, 0.2 µm polyvinylidene difluoride (PVDF) membranes and accessory utensils from Bio-Rad (Hercules, CA, USA) (Trans-Blot Turbo Transfer-pack, Cat: 1704156) were used. The SDS gels were transferred onto the PVDF membrane and blotting was performed in the Trans-Blot Turbo Transfer System (Bio-Rad, cat. no. 1704155EDU) at 25 V, 1.0 A for 30 min. After the transfer, the PVDF membrane was incubated in blocking buffer (0.2% (*w*/*v*), EZ-block, Biological industries) for 1 h and washed with TBS-T buffer (20 mM Tris, 150 mM NaCl, 0.1% Tween 20, pH = 7.4–7.6) for 5 min × 5. After washing, the membrane was incubated overnight at 4 °C in anti-FLAG antibody (ab1257, Abcam (Cambridge, UK)) diluted (1:20,000, *v*/*v*) in the blocking buffer. Following, the membrane was washed with TBS-T for 5 min × 5 and was incubated for an hour in RT in donkey anti-goat antibody (ab6885, Abcam) diluted (1:20,000, *v*/*v*) in TBS-T. The membrane was finally washed with TBS-T for 5 min × 5 and the blot was developed using Novex ECL Chemiluminescent Substrate Reagent Kit (ThermoFisher, cat. no. WP20005) by combining 0.5 mL of reagent A (containing luminol) and 0.5 mL of reagent B (containing an enhancer) to visualize the protein bands using the ChemiDoc XRS Gel imaging system (Bio-Rad). The band densitometry was performed using dedicated software (Image Lab, Bio-Rad) The SDS-PAGE gels were stained with InstantBlue Coomassie Protein Stain (Abcam, cat. no. ab119211) for 1 h on a shaking platform followed by destaining in distilled water overnight in RT. The gel images were also acquired using the ChemiDoc XRS Gel imaging system (Bio-Rad) and analyzed using dedicated Image Lab software ver. 6.1 (Bio-Rad).

### 4.13. Purification of Human β-Cardiac Myosin S1L-eGFP-FLAG Construct from Transfected C2C12 Cells

All the solutions were degassed for an hour before purification and all the steps were performed either at 4 °C or on ice in the cold room. The following description is for cells harvested from one 60 mm cell culture dish. For purification from multiple 60 mm dishes, the solutions were scaled up accordingly. The cell pellet was resuspended in 600 µL of lysis solution (20 mM imidazole pH 7.2, 100 mM NaCl, 4 mM MgCl_2_, 1 mM EDTA, 1 mM EGTA, 1 mM DTT, 3 mM MgATP, 1 mM PMSF, 10% sucrose, 0.5% Tween 20, and 1× Roche protease inhibitor cocktail) and transferred to a Dounce homogenizer and lysed with approximately 60–70 strokes on ice. The lysate was transferred into a centrifuge tube and 2 mM MgATP was added before the start of centrifugation. The lysis was centrifuged using a TLA 120.1 fixed angle rotor in Optima MAX-XP ultracentrifuge at 100,000× *g* for 1 h to separate cell remnants from the soluble protein fraction. The supernatant obtained was mixed with a purification resin consisting of agarose beads decorated with anti-FLAG antibodies (Anti-FLAG M2 affinity Gel, Sigma-Aldrich, now Merck, cat. no. A220). For one 60 mm plate of cells, 80 µL of resin slurry containing 50% resin and 50% glycerol was added to a tube. The glycerol was separated and discarded from the beads by centrifugation in Eppendorf centrifuge 5430/R fixed angle rotor (FA-45-30-11) at 2000× *g* for 30 s. The beads were equilibrated by resuspending them in 600 µL of lysis solution and centrifuging at 2000× *g* for 30 s. This step was repeated three times. The supernatant obtained after ultracentrifugation (cell lysate) was added to the equilibrated resin (40 µL packed resin) and nutated in a cold room for 1 h. After incubation, the resin was spun down at 2000× *g* for 2 min and was washed twice with 20-25 times the resin volume with wash buffer (20 mM Imidazole pH 7.2, 150 mM NaCl, 5 mM MgCl_2_, 1 mM EDTA, 1 mM EGTA, 1 mM DTT, 3 mM MgATP, 10% sucrose, 1×  complete Mini Protease protease inhibitor cocktail) followed by two more washes without the addition of MgATP and protease inhibitor cocktail. For each wash, the beads were nutated in wash buffer for 10 min followed by centrifugation at 2000× *g* for 2 min. After the last wash, ~2 µL of the resin was removed with a cut pipette tip and placed on a 24 × 60 mm^2^ coverslip glass. A volume of 10 µL of wash buffer was added and an 18 × 18 mm^2^ coverslip glass was placed on top. The resin was observed under a fluorescence microscope (see above) using a FITC filter set and 40X objective magnification. The resin with bound proteins was resuspended in 200 µL elution buffer (20 mM Imidazole pH 7.2, 150 mM NaCl, 5 mM MgCl_2_, 1 mM EDTA, 1 mM EGTA, 1 mM DTT, 10% Sucrose, and 150 μg/mL of 3 × FLAG peptide [Sigma-Aldrich, now Merck, cat. No. F4799)]) and incubated for 20 min. The resin was spun down at 2000× *g* for 2 min and the eluate was filtered through Pierce Micro-Spin columns (Thermo Scientific, cat. no. 89879) to filter out any remaining resin at 10,000× *g* for 1 min. If necessary, the proteins were also concentrated using Amicon Ultra centrifugal filter units, 50 kDa MWCO (Merck, cat. no. UFC505024). To determine if the bound protein was completely eluted from the purification beads, a volume of ~2 µL of the resin was taken and put on top of a glass slide and imaged under the fluorescence microscope (as described above). After purification, 1 mg/mL BSA (final concentration) was added to the purified proteins and dialyzed three times (first for 1 h, followed by overnight, and finally for another hour) against 125 mL of X-linking wash buffer, pH 7.5 (10 mM imidazole, 5 mM KCl, and 3 mM MgCl_2_, 1 mM DTT) as described previously [28]. Note that protein dialysis was performed only for single-molecule basal and actin-activated ATPase assay.

### 4.14. Concentration Estimation of Purified S1L-eGFP-FLAG Protein

The concentration of the purified S1L-eGFP-FLAG protein was determined using a Tecan microplate reader (Spark, Berkeley, CA, USA) controlled by dedicated software (SparkControl Ver 3.2). A standard curve containing 5 different concentrations of eGFP (MyBioSource (San Diego, CA, USA), Cat. MBS146512) diluted in 50 µL X-linking wash buffer was run in parallel on the 96-well microplate (Greiner Bio-one (Kremsmünster, Austria), cat. no. 655077), together with 50 µL of the purified protein (in X-linking wash buffer). The standard curve concentration range was prepared to approximately match the expected concentration of S1L-eGFP-FLAG purified from one or more 60 mm dishes, usually in a range 5–20 nM. If a significant number of cells were used for protein purification, the purified protein sample was diluted (if necessary) to fit between these data points. A total volume of 50 µL solution was loaded in each well. For eGFP fluorescence reading, top measuring mode was used with excitation and emission wavelengths set to 485/20 nm and 535/20 nm, respectively, as an average of 5 flashes/well. The software calculated and set the gain from the most concentrated eGFP sample of the standard curve, as well as the z-position from the middle eGFP sample of the standard curve. The eGFP and sample values were blanked against an X-linking wash buffer. Sample concentration was extrapolated from the standard curve via linear regression. Any dilution factor was taken into consideration.

### 4.15. Single-Molecule Basal ATPase Assay

Single-molecule basal ATPase was performed as described in the previous publications [28,29] using total internal reflection fluorescence (TIRF) microscopy. In brief, the TIRF assay solution was prepared a day before the experiment, transferred to a syringe, and stored on ice overnight. Trolox-Trolox/Trolox-Quinone LISS (TX/TQ-LISS) was prepared by dissolving Trolox in methanol at a concentration of 100 mM and subsequently diluting it in LISS to a concentration of ~2 mM. The pH was adjusted to 7.4 using 1 M KOH. The solution was sterile filtered (0.2 µm) into a petri dish and exposed to UV light (254 nm) at 120,000 µJ/cm^2^ for 15 min using Stratalinker1800 (Stratagene, San Diego, CA, USA). Following this, the solution was degassed and then used to prepare the assay solution. TIRF assay solution (ionic strength–60 mM) was prepared in TX/TQ-LISS containing 45 mM KCl, 10 mM DTT, 7.2 mg/mL glucose, 3 U/mL POX, 0.01 mg/mL catalase, 2.5 mM CP, 0.2 mg/mL CPK, 2 mM COT, 2 mM NBA, 5 nM Alexa647–ATP, and 0.64% methylcellulose. Wash buffer was prepared by degassing Low Ionic Strength Solution (LISS, 10 mM MOPS, pH = 7.4 at 25 °C, 1 mM MgCl_2_, 0.1 mM EGTA) and supplementing with 50 mM KCl and 1 mM DTT.

For TIRF-based assays, high-precision cover glasses (24 × 50 mm^2^, 170 ± 5 µm thickness, No. 1.5 H, ThorLabs (Newton, NJ, USA), cat. no. CG15KH1) were used. Cover glasses were surface cleaned using air plasma ashing (plasma cleaner Femto Standard Diener electronic GmbH, Ebhausen, Germany) at 100 W (40 kHz) and 0.6–0.8 mbar pressure for 3 min. A flow cell was prepared on the clean side of the cover glass by coating it with 1% nitrocellulose in amyl acetate, air dried, and topped with an 18 × 18 mm^2^ coverslip using a double-sided adhesive tape (3M-Scotch, Saint Paul, MN, USA) as described previously [28]. For the assay, the flow cell was first coated with anti-GFP antibodies (Merck, cat. no. MAB3580) diluted 10 times in wash buffer, incubated for 2 min followed by the addition of 1 mg/mL BSA (2 min). The chamber was then washed twice with wash buffer, followed by the addition of purified functional S1L-eGFP-FLAG myosin construct (diluted to 1/200–1/500 in wash buffer containing 0.1 mg/mL BSA), if standard purification was performed using cells from a single 60 mm cell culture dish for 5 min (otherwise the dilution factor needed to be adjusted). The unattached myosin was removed by washing the chamber twice with wash buffer, followed by the addition of 100 nM ATP for 2 min. The chamber was rinsed thrice with wash buffer before the addition of the TIRF assay solution. The binding of Alexa647-ATP to purified S1L was observed using our in-house built TIRF microscope equipped with a 60× oil immersion objective, NA 1.49, Cy5 filter cube, and a red laser (Melles Griot 05-LHP-925 30 mW, 632.8 nm HeNe laser). A blue laser (Changchun New Industries Optoelectronics Tech. Co., Ltd. (Changchun, China), Blue Solid-State Laser, MLL473, 473 nm, 50 mW) and a FITC filter cube were used to observe the eGFP signal. Videos were recorded for 15 min at 23 ± 1 °C using an EMCCD camera Andor iXon Ultra 897 controlled by NIS Elements software (Nikon (Tokyo, Japan), ver. 4.51) with gain and exposure time set at 100 and 50 ms, respectively. The entire TIRF microscopy setup is described in depth elsewhere [29]. Acquired videos were pre-processed by converting them from 16- to 8-bit frame depth, followed by background subtraction using the ImageJ function (Process/Subtract Background), setting the rolling ball radius to 5 pixels. The time projection of the Alexa647 signal (generated by ImageJ function Image/Stack/Z project/Standard deviation) and time projection of the eGFP signal (generated by ImageJ function Image/Stack/Z project/Average) were merged by using ImageJ function (Image/Color/Merge channels). Colocalizations of Alexa647–ATP and myosin eGFP were chosen as the hotspots from which Alexa647 fluorescence traces were extracted using the ImageJ function (Image/Stacks/Plot *Z*-axis Profile). The dwell-on times were collected from the extracted traces using a custom-made MATLAB script [29] and cumulative dwell time distributions were plotted as described previously [29,52] and were then fitted to double-exponential functions using non-linear regression in GraphPad Prism (v.9.4.1). Rate constants and amplitudes were obtained from the fit and are represented as mean ± 95% confidence intervals.

The fraction of active heads was estimated with myosin constructs immobilized to surfaces via anti-GFP antibodies based on the fractional co-localization of eGFP and Alexa647.

### 4.16. Single-Molecule Actin-Activated ATPase Assay

Cross-linking of expressed β-cardiac myosin was performed according to our recent publication [28]. Briefly, 5 µL of the S1L-eGFP-FLAG myosin protein (~13 nM from JetPrime and 20 nM from GenJet expression system) purified from a single 60 mm cell culture dish of expressing cells was added to a chamber with cross-linked phalloidin stabilized F-actin filaments cross-linked to the surface [28]. Upon observed binding of myosin to F-actin in the rigor state, the cross-linking was initiated by introducing 15 mM EDC in MES buffer (50 mM MES pH 6.5, 1 mM DTT) to the chamber for 10 min. A series of washes were performed with X-linking wash buffer and in addition with X-linking wash buffer supplemented with 50 nM ATP and 0.5 mg/mL BSA to remove non-cross-linked heads together with blocking non-functional motor heads. Another sequence of washing with X-linking wash buffer was performed, and finally, two droplets of assay solution were added to the chamber. The experiments were performed at 23 ± 1 °C. The TIRF assay solution for actin-activated assays was modified from the one used for basal ATPase assay to contain 500 nM nonfluorescent ATP, together with 0.1 mg/mL BSA [28]. For data analysis, colocalized Alexa647 and eGFP fluorescence areas located on the F-actin filaments were taken into consideration. Co-localization was assumed if Alexa647 fluorescence was detected in pixels representing the filaments (as suggested by the S1L-GFP labeling). The video processing, fluorescence trace extraction, and dwell collections follow the ones described above for basal ATPase assays and are described in detail elsewhere [28]. The cumulative dwell time distributions were plotted as described previously and were then fitted to the sum of exponential functions using non-linear regression in GraphPad Prism (v.9.4.1). Rate constants and amplitudes were obtained from the fit and are represented as mean ± 95% confidence intervals.

Some S1L-eGFP-FLAG constructs non-specifically adsorb to the nitrocellulose outside the actin filament rather than binding to the actin filament when myosin is added during the cross-linking procedure [28]. One may consider using these myosin motor domains to estimate basal ATP turnover rate. However, they may be adsorbed in inappropriate ways, e.g., via their actin-binding end rather than via GFP because anti-GFP antibodies are lacking in the experiments with actin-cross-linked myosin. Therefore, it may not be reliable to estimate basal myosin ATPase because adsorption via the actin-binding region is likely to modify or even inhibit the ATPase (cf. [53,54]).

### 4.17. In Vitro Motility Assays

The sliding velocity of actin filaments propelled by purified S1L-eGFP-FLAG protein was determined using standard in vitro motility assay (IVMA) as described earlier [27]. Initially, the purified proteins were subjected to the removal of non-functional motor heads (“deadheading”) by performing affinity purification (twice). For this, the S1L-eGFP-FLAG elution was mixed with non-fluorescent F-actin at a concentration 10 times that of the purified protein. This mixture was then incubated on ice for 5 min, followed by the addition of 4 mM MgATP and further incubation on ice for 5 min. Subsequently, the mixture was centrifuged at 244,990× *g* (75,000 RPM) for 10 min at 4 °C using TLA 120.1 rotor in Optima MAX-XP centrifuge (Beckman Coulter, Brea, CA, USA). The supernatant containing functional motor heads was directly used for the assay. Wash buffer was prepared by degassing LISS and supplementing the solution with 50 mM KCl and 1 mM DTT. The motility assay solution was prepared a day before the experiment, transferred to a syringe, and stored on ice overnight [55]. Assay solution (ionic strength-60 mM) contains 45 mM KCl, 2 mM MgCl_2_, 10 mM DTT, 0.64% methylcellulose, 2 mM MgATP, along with an oxygen scavenging system (3 mg/mL Glucose, 40 µg/mL catalase, 100 µg/mL glucose oxidase (GOX)) and ATP regeneration system (200 µg/mL CPK and 2.5 mM CP). A standard flow cell was prepared by coating the surface of a 24 × 60 mm^2^ coverslip (#0, Menzel-Gläser, Braunschweig, Germany) with 1% nitrocellulose in amyl acetate prepared by mixing an equal amount of 2% collodion (parlodion) in amyl acetate (Electron Microscopy Science, Hatfield, PA, USA, Ref: 12620-50) with amyl acetate (Electron Microscope Science, Ref: 10816), air dried, and topped with 18 × 18 mm^2^ coverslip using a double-sided adhesive tape (3M-Scotch). The anti-GFP antibodies (Merck, cat. no. MAB3580), diluted 6 times in wash buffer, were attached to the prepared coverslips by incubating them for 2 min followed by the addition of 1 mg/mL BSA (2 min). The chamber was then washed once with wash buffer, followed by the addition of purified functional S1L-eGFP-FLAG myosin construct (final concentration after deadheading ~125 nM) for 5 min. Any residual non-functional myosin molecules were further blocked by 1 µM non-fluorescent F-actin (“blocking actin”) for 2 min. The blocking actin was prepared by shredding the actin filaments ~200 times in wash buffer using a syringe/pipette [56]. This was followed by the addition of 2 mM MgATP (2 min). The chamber was washed twice/three times with wash buffer followed by the addition of 15 nM rhodamine–phalloidin-labeled F-actin (2 min). Finally, the IVMA assay solution was added. Actin filament sliding was recorded using an EMCCD camera (C9100-12, Hamamatsu photonics, controlled by HCImage software) under 63X objective (fitted with an objective heater) and Cy3 filter set of Zeiss Axio Observer epifluorescence microscope at 25 ± 1 °C with and at 5 frames/s (16-bit depth). A custom-made MATLAB program was used to calculate the sliding velocities [57,58] and Fiji to determine the fraction of motile filaments by dividing motile filaments by total filaments per video recorded.

## 5. Conclusions

We have demonstrated the usefulness of the alternative transfection reagent GenJet for the transfection of C2C12 cells with the plasmid-carrying gene for the β-MHC for subsequent expression and purification of functionally active protein. We have developed an optimal protocol for the use of GenJet and demonstrated some potential advantages over the use of JetPrime, particularly for the expression of large quantities of a given MHC variant. Our results suggest that GenJet can substitute JetPrime for virus-free C2C12 cell transfection for expression and purification of active vertebrate striated muscle myosin II motor proteins. This supports the sustainability of the virus-free transfection method. Presently, however, the MHC production using JetPrime is better established calling for more studies before routine use of GenJet. On a more general level, we foresee that virus-free transfection methods, together with miniaturized assays as exemplified here, enable more wide-spread use (beyond the few labs active in the field today) of reverse genetics studies of striated muscle myosin. These studies will also be more readily expanded to a wide range of systematically selected mutations benefitting both insights into the HCM pathogenesis and fundamental understanding of actomyosin operation in striated muscle. Moreover, finally, our study demonstrates the possibility to use two different non-viral transfection reagents to express a rather complex protein construct in C2C12 cell lines, known to be rather hard to transfect [27]. This hints towards the likelihood to express a wider range of complex, functional proteins in different mammalian cell lines.

## Figures and Tables

**Figure 1 ijms-25-06747-f001:**
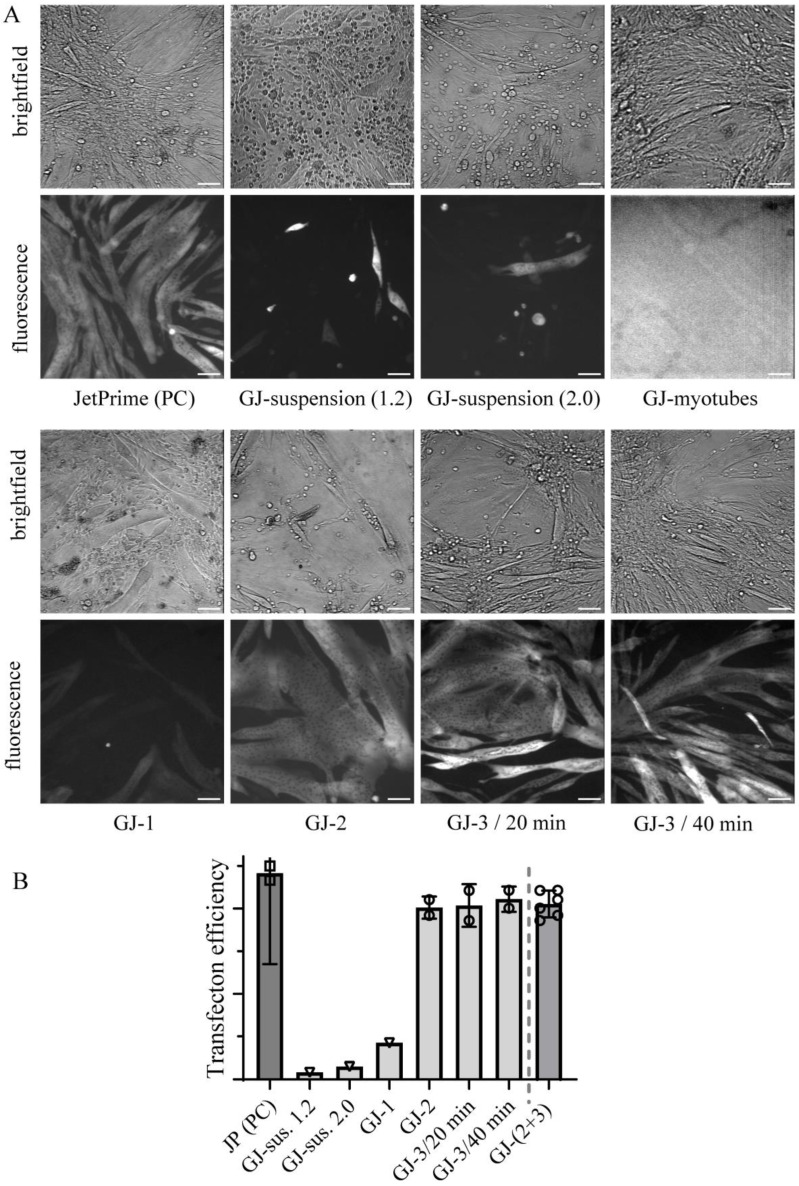
GenJet transfection efficiency at different transfection conditions. (**A**) Representative brightfield and fluorescence microscopy images of cells plated in 35 mm dishes at different transfection conditions (see also Table 1) acquired on day 7 post-transfection (except transfection of 1.2 × 10^6^ cells in suspension, where the images were taken on day 5 post-transfection) and compared to JetPrime positive control (PC). Note poor differentiation capabilities of cells transfected in suspension regardless of cell number used (1.2 × 10^6^ or 2.0 × 10^6^) and days post-transfection (5 or 7) with a substantial number of rounded cells. No fluorescent cells were observed when transfecting fully differentiated myotubes, even with a fluorescence image taken at overexposure settings. Bars represent 100 µm. (**B**) Quantification of transfection efficiencies on fluorescence images (see Section 4 and Appendix A for details) and comparison to JetPrime (JP positive control (PC, squares). Approaches including transfecting cells in suspension and using protocol GJ-1 (inverse triangles) performed the worst. There was no significant difference between the data under GenJet transfection conditions GJ-2, GJ-3/20 min, and GJ-3/40 min (circles), motivating that these be pooled into one group (last data point to the right; GJ-2+3, circles). Data points per condition represent independent experiments performed on different days and are calculated as the average value from at least 5 randomly acquired fluorescent images (field of views) per 35 mm cell culture dish. Where applicable, mean ± 95% confidence intervals (error bars) were calculated.

**Figure 2 ijms-25-06747-f002:**
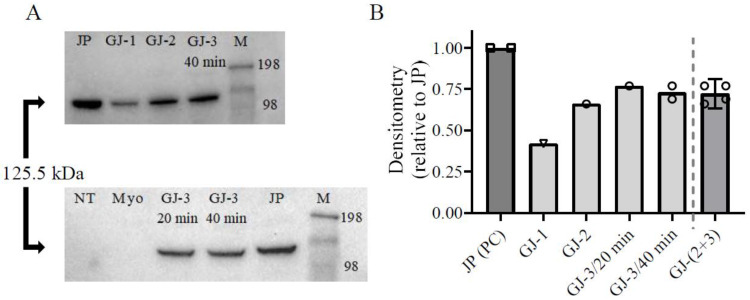
Expression analysis of total cell lysates of transfected cells in different transfection conditions. (**A**) Western blots using anti-FLAG antibodies to quantify the amount of expressed 125.5 kDa S1L-eGFP-FLAG construct. The original blots are presented in Appendix A. JP: cells transfected with JetPrime as a positive control (PC). NT: non-transfected cells (negative control), Myo: transfection of fully differentiated myotubes, M: molecular weight ladder (kDa). For other GenJet conditions (GJ), please see Table 1. (**B**) Densitometry analysis of the blots under (**A**). Transfection condition GJ-1 (inverse triangle) performed the worst in comparison to positive control (PC, squares). Note the minor difference between expression levels under GenJet transfection conditions GJ-2, GJ-3/20 min, and GJ-3/40 min (circles) motivating pooling of these into one group (last column to the right; GJ-2+3, circles). Where applicable, mean ± 95% CI (error bars) are given.

**Figure 3 ijms-25-06747-f003:**
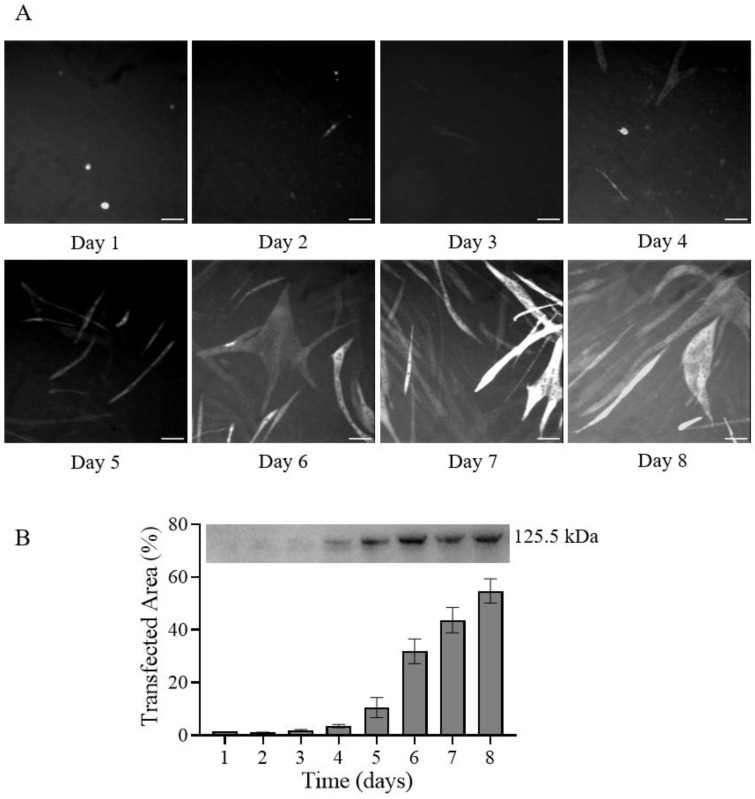
Transfection efficiency at each of 8 days post-transfection. (**A**) Representative fluorescence microscopy images of cells plated in 35 mm dishes and transfected using the GJ-3/20min protocol (Table 1) acquired on different days post-transfection. Scale bars represent 100 µm. (**B**) Quantification of transfection efficiency over time as a fraction of area with GFP-fluorescence (**bottom**) and Western blot using anti-FLAG antibodies to quantify the amount of expressed 125.5 kDa S1L-eGFP-FLAG construct in total cell lysates (**up**). The original blot is presented in Appendix A. Bars represent mean ± SD of 5 images (field of views) per cell culture dish.

**Figure 4 ijms-25-06747-f004:**
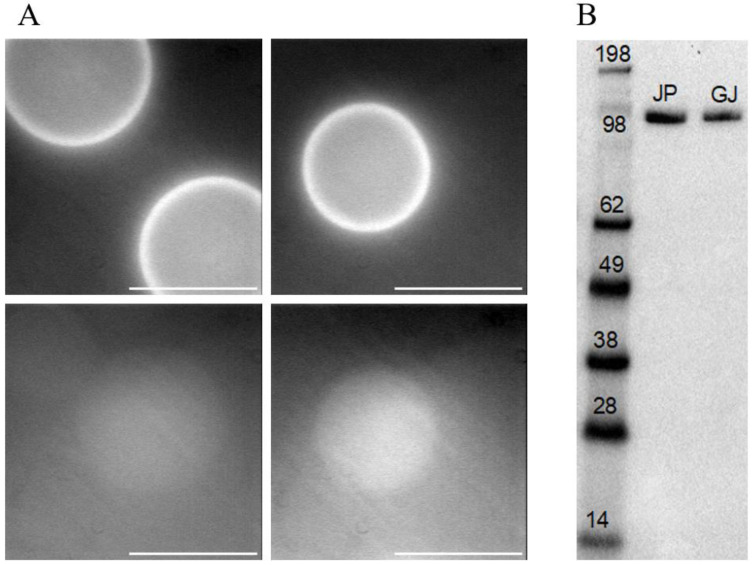
Affinity protein purification using anti-FLAG antibody resin (beads). (**A**) (**Top**): GFP fluorescence of beads after capturing the expressed protein S1L-eGFP-FLAG acquired with the camera gain 150 and exposure time 100 ms. (**Bottom**): Beads (acquired with the camera gain 150 and exposure time 300 ms) showing the loss of fluorescence after elution with 3× FLAG peptide. Scale bars: 100 µm. (**B**) Western blot showing purified human β-cardiac myosin heavy chain S1L-eGFP-FLAG construct from one 60 mm cell culture dish of cells transfected either with JetPrime (JP) as positive control or GenJet (condition GJ-3/20min, Table 1, scaled up for 60 mm plate). The comparative densitometry analysis showed the GJ purification yield to be 0.78 of the JP. The original blot is presented in Appendix A.

**Figure 5 ijms-25-06747-f005:**
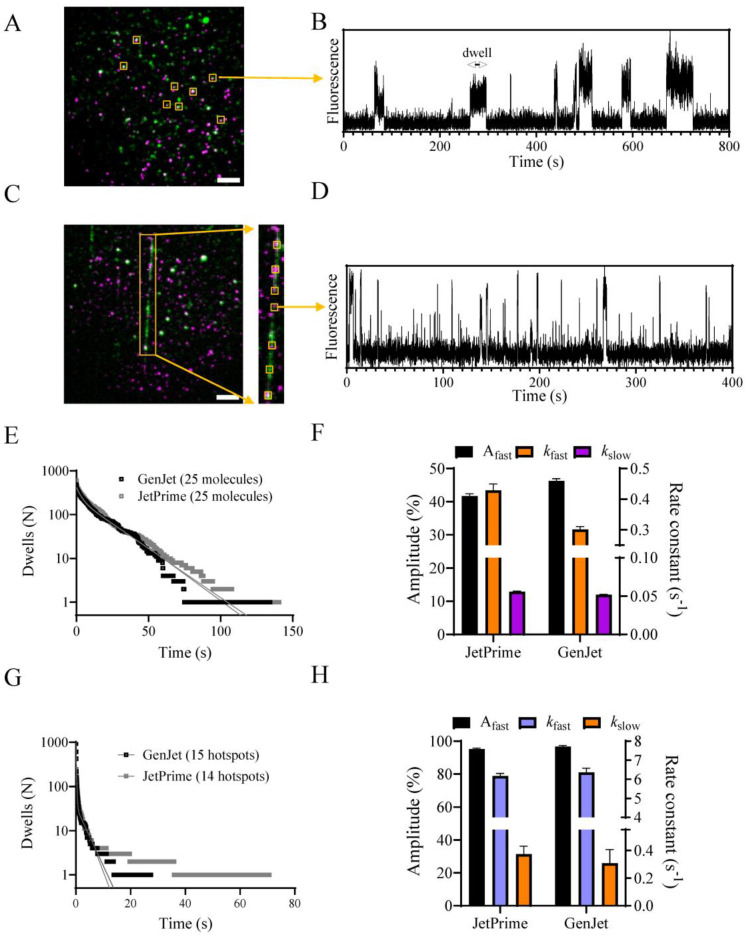
Single-molecule ATPase using purified S1L-eGFP-FLAG myosin construct and Alexa647-ATP. (**A**) Merged images of eGFP fluorescence (average time projection of 10 frames at rate 20 s^−1^) in green and Alexa647–ATP fluorescence in magenta (standard deviation time projection of 800 s video, acquired at 20 s^−1^) obtained using total internal reflection fluorescence (TIRF) microscopy. The yellow squares represent examples of colocalization of both signals from where the time fluorescence traces were extracted. The myosin construct expressed and purified from cells transfected with GenJet (presented here) or JetPrime (positive control) were attached to the surface via anti-GFP monoclonal antibodies deposited on nitrocellulose-coated glass slides. Scale bar: 5 µm. (**B**) Representative Alexa647 fluorescence trace of single-molecule basal myosin ATPase depicting characteristic binding events (dwells) from which basal myosin ATPase rate constant was estimated. (**C**) Merged images of eGFP fluorescence (average time projection of 10 frames at rate 20 s^−1^) in green and Alexa647–ATP fluorescence in magenta (standard deviation time projection of 400 s video, acquired at 20 s^−1^) obtained using total internal reflection fluorescence (TIRF) microscopy. The myosin construct expressed and purified from cells transfected with GenJet (presented here) or JetPrime (positive control) were cross-linked to surface-immobilized actin filaments. The zoom-in image detail highlights one such actin filament showing colocalizing eGFP and Alexa647 fluorescence. The yellow squares represent examples of the hotspots on actin filaments from where the time fluorescence traces were extracted. Scale bar: 5 µm. (**D**) Representative Alexa647 fluorescence trace of single molecule actin-activated myosin ATPase depicting characteristic short binding events from which actin-activated myosin ATPase rate constant was estimated. (**E**) Cumulative frequency distribution of Alexa647–nucleotide dwell time events obtained from colocalized surface hotspots were fitted with double-exponential functions (solid lines). GenJet data from 25 myosin molecules, N_dwell_ = 488, JetPrime data from 25 myosin molecules, N_dwell_ = 605. (**F**) Amplitudes and rate constants from the fitting of data in C, where the fast rate constant was attributed to unspecific Alexa647–ATP binding to myosin molecules (*k*_unspecific_), while the slow rate constant was attributed to basal myosin ATPase activity (*k*_basal_). (**G**) Cumulative frequency distributions of Alexa–nucleotide dwell time events comparing actin-activated ATPase activity from myosin construct purified from cells transfected with GenJet or JetPrime. GenJet data from 15 actomyosin hotspots, N_dwell_ = 949, JetPrime data from 14 actomyosin hotspots, N_dwell_ = 936. The actomyosin data were fitted with double-exponential functions. Note that an alternative fit is possible for JetPrime data (see Appendix A). (**H**) Amplitudes and rate constants from the fitting of data in G, where the fast rate constant was attributed to actin-activated myosin ATPase activity (*k*_cat_), while the slow rate constant was attributed to unspecific Alexa647–ATP binding to myosin molecules (*k*_unspecific_). Error estimates refer to 95% confidence intervals derived in the regression analysis. Temperature: 23 °C.

**Figure 6 ijms-25-06747-f006:**
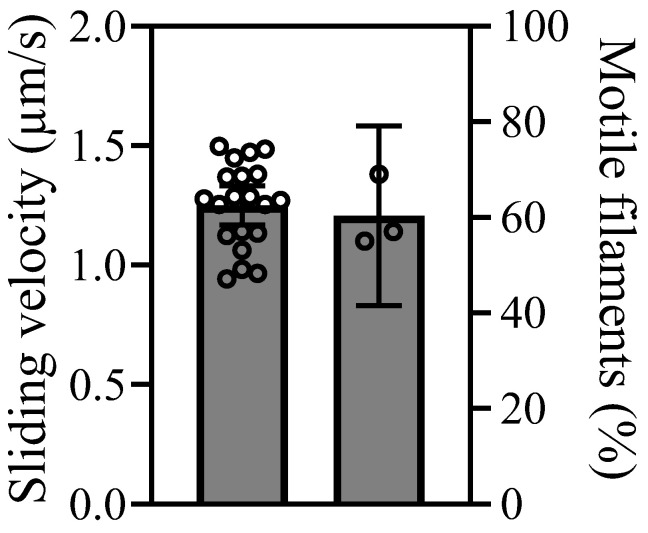
In vitro motility assay (IVMA) for S1L-eGFP-FLAG myosin protein purified from cells transfected using GenJet. The sliding velocities of actin filaments are superimposed on mean ± 95% CI from 20 individual filaments (circles, left). The fraction of motile filaments (FMF) after two affinity purifications (“deadheadings”) is given to the right. The data are superimposed on mean ± 95% CI from three fields of view (circles, right) in the flow cell studied. Temperature: 25 °C.

**Figure 7 ijms-25-06747-f007:**
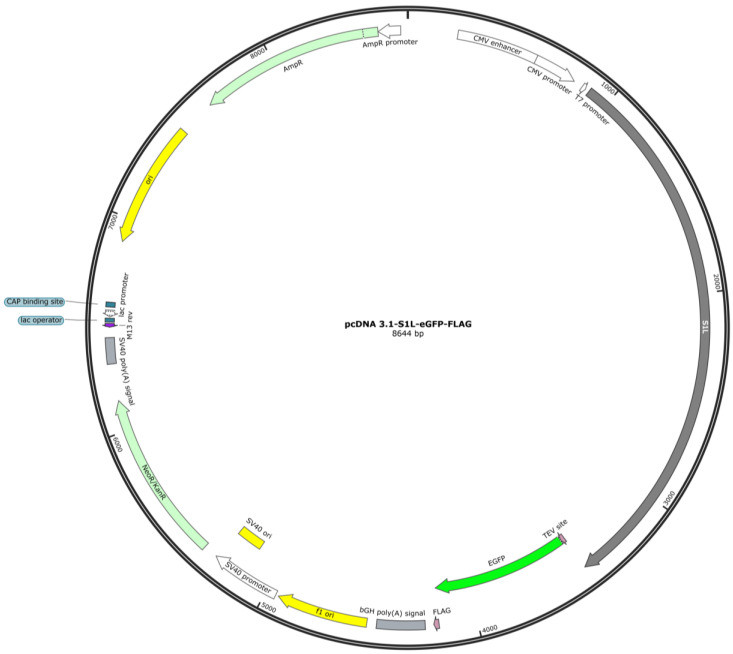
Schematic of plasmid used for transfection of C2C12 cells. S1L: S1-long corresponding to the N-terminal of the β-MHC (residues 1-848). TEV: tobacco etch virus. EGFP: enhanced green fluorescent protein. FLAG: flag peptide for affinity-based purification. Figure prepared using software SnapGene (ver 6.0.2).

**Table 1 ijms-25-06747-t001:** Different conditions for transfecting adherent and confluent C2C12 cells growing in a 35 or 60 mm cell culture dish using the transfection reagent GenJet (GJ).

Condition, Plate Size, Complex Formation Volume	Plasmid Dilution for Transfection Complex Formation	GenJet Dilution for Transfection Complex Formation	Final Transfection Volume Complex Dilution Ratio	Cell Transfection Time
^a^ GJ-1: 35 mm, 210 µL	2 µL plasmid +100 µL DMEM	8 µL GenJet +100 µL DMEM	210 µL 1:20	20 min
GJ-2: 35 mm, 420 µL	4 µL plasmid +200 µL DMEM	16 µL GenJet +200 µL DMEM	420 µL 1:20	20 min
GJ-3: 35 mm, 210 µL	2 µL plasmid +100 µL DMEM	8 µL GenJet +100 µL DMEM	^b^ 410 µL 1:40	20 or 40 min
GJ-3: 60 mm, 525 µL	5 µL plasmid +250 µL DMEM	20 µL GenJet +250 µL DMEM	^c^ 1025 µL 1:40	20 min

^a^ used also for the transfection of cells in suspension and myotubes (from official manuals). ^b^ 200 µL DMEM was added to the cells before the addition of the transfection complexes of 210 µL. ^c^ 500 µL DMEM was added to the cells before the addition of the transfection complexes of 525 µL.

**Table 2 ijms-25-06747-t002:** Summary of ATPase and IVMA results.

Parameters (Mean ± SEM)	S1L-eGFP-FLAG GenJet Transfected Cells		S1L-eGFP-FLAG JetPrime Transfected Cells	
*k*_basal_ ^§^ A_basal_ ^§^	0.052 ± 0.0003 s^−1^ 53.7 ± 0.5%		0.056 ± 0.0003 s^−1^ 58.3 ± 0.4%	
*k*_cat_ ^§^ A_cat_ ^§^	6.4 ± 0.1 s^−1^ 96.7 ± 0.4%		6.2 ± 0.1 s^−1^ 95.2 ± 0.3%	
*k*_unspecific_ ^§^ A_unspecific_ ^§^	^a^ 0.30 ± 0.01 s^−1^ ^a^ 46.3 ± 0.4%	^b^ 0.31 ± 0.04 s^−1^ ^b^ 3.3 ± 0.01%	^a^ 0.43 ± 0.01 s^−1^ ^a^ 41.7 ± 0.3%	^b^ 0.37 ± 0.02 s^−1^ ^b^ 4.8 ± 0.02%
Actin filament velocity (25 °C, n = 20 filaments)	1251 ± 339 nm/s		^c^ 1823 ± 63 nm/s	
Fraction of motile filaments (25 °C, n = 3 fields of view)	60 ± 4%		^c^ 69 ± 4%	

^§^ rate constants, amplitudes, and errors are derived in the fitting process of double exponential function to cumulative distributions of single-molecule data. T = 23 °C. One protein preparation studied for both GenJet and JetPrime. ^a^ estimated from single-molecule basal ATPase assay. ^b^ estimated from single-molecule actin-activated ATPase assay. ^c^ from reference [27], performed at 25 °C after “double-deadheading”.

**Table 3 ijms-25-06747-t003:** The price performance of GenJet (GJ) and JetPrime (JP) transfection reagents.

	JetPrime ^a^	GenJet
Price/µL	^b^ SEK 8.43	^c^ SEK 1.87
Price/60 mm plate	SEK 189.75	SEK 37.4
Protein yield (relative to JP)	100%	>70%
Price/S1L (60 mm plates)	SEK 189.75	SEK 53.43

^a^ assuming transfections are performed according to our optimized protocol [27]. ^b^ calculated from the JetPrime reagent price (without tax) from Bionordika (Sweden, quote date 11 April 2024). ^c^ calculated from the GenJet reagent price from Tebu-bio (Denmark, webpage price on 15 April 2024).

**Table 4 ijms-25-06747-t004:** Optimized number of seeded cells to obtain confluence in different times based on our studies.

The Desired Number of Cells × 10^3^	Estimated Time to Reach 60–70% Confluency (Desired Confluence)
1	3 days
0.5	4 days

## Data Availability

Most data generated or analyzed during this study are included in this published article (and its Appendix A). Additional data generated during or analyzed during the current study are available from the corresponding authors on reasonable request. The DNA sequence encoding human beta cardiac myosin motor domain (S1L) generated during the current study is available in the NCBI GenBank depository (part of International Nucleotide Sequence Collaboration [INSDC]) under accession number OQ092356.

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
