# Peer review of "Cost-Efficient Expression of Human Cardiac Myosin Heavy Chain in C2C12 Cells with a Non-Viral Transfection Reagent"

_ijms, 2024, doi:10.3390/ijms25126747_

Round 1
Reviewer 1 Report
Comments and Suggestions for Authors
This paper presents an important cost-effective improvement for the production of recombinant full-length human cardiac myosin. It is an important development that will help many research groups to produce such myosin, thereby significantly boosting the field. The production of human cardiac myosin is a very important issue and a very important hot topic for decades in the field of basic research and also in drug development as drug development targeting myosin is one of the most important innovations filed at the moment.
The paper is logical and very well presented. There is one small question: in Fig. 3b the authors show the amount of myosin expressed as a function of time. It seems that on day 8 the expression level is still not saturated, so one might ask why the authors did not wait longer for saturation.
Because of the practical importance and the high quality of the writing, I recommend this paper for publication as is.
Author Response
Please see the attachment that includes responses to all reviewers

Reviewer 2 Report
Comments and Suggestions for Authors
The paper describes the protocol for the expression of human cardiac β-myosin heavy chain in C2C12 cells without the standard adenovirus-based method used in a limited number of laboratories. The possibility of exploiting non-adenoviral vectors is rather crucial for research in the field, as would increase the accessibility of myosin expression constructs for a number of studies regarding the physiology and pathology of striated muscle. With respect to a previous attempt by the same group using another non-viral agent, the authors show that the transfection reagent used in this study (GenJet) allows a slightly lower yield but still offers a reasonable amount of functional protein at lower costs.
The overall quality of the manuscript is good and, most importantly for a methodological paper, the experimental procedures are in general described with sufficient detail both in the main text and in the Materials and Methods section.
I do have some questions and minor concerns:
1) If the transfection efficiency values (Fig. 1B) and the densitometry values (Fig. 2B) of conditions GJ-2 and GJ-3 are not significantly different so that it is reasonable to pool them into one group (GJ-(2+3)), the p-value of the differences should be reported.
2) In the text you report the final yield as relative to that of JetPrime transfection system. I think the absolute value (µg/culture dish) should be mentioned explicitly at the end of the purification step.
3) In the zoom-in-image of Fig. 5C, you select an actin filament with a number of hotspots showing colocalization of eGFP-myosin and Alexa647-ATP fluorescence, from which time fluorescence traces are extracted for the estimate of actin-activated ATPase activity. Do you think it’s reasonable to expect to estimate myosin basal activity from the regions outside the strip defined by the actin filament geometry, or do you presume to have only actin-crosslinked myosins on the cover glass? In the latter case, what is then the filament length/width range that you consider as valid for hotspots selection? Could you explain more in detail this aspect of your measurements throughout the main text?
4) At lines 429-452, you discuss about the gliding velocity of actin filaments in IVMA being lower than that obtained with the JetPrime transfection approach (1251 ± 339 nm/s vs 1823 ± 63 nm/s): is this difference significant? However, in your previous paper (Velayuthan et al. 2023) S1L-eGFP-FLAG was co-purified with ELCs and RLCs, while there is no mention of this step in this case either in the text or in Fig. 4B, so I presume that the gliding velocity was measured from the S1L-eGFP-FLAG construct alone. Is there a reason behind the choice of missing the co-purification step? Considering the modulation of the light chains on motile and ATPase kinetic properties (Amrute-Nayak et al. Small 2019, Osten et al. J Gen Physiol 2022 just as examples), could this also be a possible source of difference for the actin velocities of GenJet and JetPrime S1 constructs?
5) It would be interesting to comment on the possibility to use the same expression method for a longer, HMM-like, construct.
The following typos should be amended throughout the text:
Line 98: JetPrime is misspelled
Lines 107 and 169: I think “that” should be replaced by “those”
Line 154: myoblasts
Figure 1B: correct y-axis label (“Transfecton”)
Line 280: kcat has an additional character
Lines 283-284: I presume “Figure 5” should be eliminated from this sentence
Lines 436, 753 and 810: “)” is missing
Author Response

(The authors gave the same response as above.)

Reviewer 3 Report
Comments and Suggestions for Authors
The authors aimed to report that GenJet is a low-cost reagent for large-scale vertebrate muscle myosin production in C2C12 cells.
.
They improved GenJet by modification using the protocol of JetPrime to achieve the low cost culture of C2C12 cells. They examined the protein purification yields and verified the enzymatic activity of the construct using an in-vitro motility assay and single molecule fluorescence-based ATPase assays and confirmed that the purified myosin construct is active.
.
The experiments were well conducted. However, I raised a few comments.
1) The authors improve the virus-free infection and expression of myosin gene to C2C12 cells by using JetPrime. However, this improvement is applicable to the virus-free infection of other genes to the other cells. If it can be generalized, this method will provide benefits for a wide range of researchers. Please comment or show the data in the text.
2) They measured basal ATPase rate of myosin on the glass and actin activated maximum ATPase rate of myosin EDC-crosslinked to F-actin from single molecule fluorescence of AlexaATP. If some damaged myosin with eGFP fluorescence does not bind AlexaATP, they might ignore such damaged myosin molecules under fluorescence microscope. Please comment in the text for how many % of damaged myosin exist in purified sample.
3) Furthermore, they did not measure the affinity of actin and myosin during ATPase and Kactin of actomyosin ATPase rate. The actin affinity is easily impaired (weakened or permanently bound) by SH modification etc. Please comment in the text.
4) In vitro motility assay, small difference in the velocity and motile faction between GenJet and Jetprime might be due to the difference in the fraction of damaged myosin related to points 2 and 3. Please comment in the text.
Author Response

(The authors gave the same response as above.)
